# *Pinus* spp. Somatic Embryo Conversion under High Temperature: Effect on the Morphological and Physiological Characteristics of Plantlets

Antonia Maiara Marques do Nascimento [1], Priscila Alves Barroso [2],
Naysa Flavia Ferreira do Nascimento [3], Tomás Goicoa [4,5], María Dolores Ugarte [4,5],
Itziar Aurora Montalbán [1,*] and Paloma Moncaleán [1,*]

[1] Departamento de Ciencias Forestales, Neiker-BRTA, 01080 Arkaute, Spain; mmarques@neiker.eus
[2] Departamento de Agronomía, Campus Professora Cinobelina Elvas, Universidade Federal do Piauí, Bom Jesus, Piauí 64900-000, Brazil; pa.barroso@hotmail.com
[3] Departamento de Fitotecnia e Ciências Ambientais, Universidade Federal da Paraíba, Areia, Paraíba 58397-000, Brazil; naysafn@gmail.com
[4] Departamento de Estadística, Informática y Matemáticas, Universidad Pública de Navarra, 31006 Pamplona, Spain; tomas.goicoa@unavarra.es (T.G.); lola@unavarra.es (M.D.U.)
[5] INMAT², Universidad Pública de Navarra, 31006 Pamplona, Spain
[*] Correspondence: imontalban@neiker.eus (I.A.M.); pmoncalean@neiker.eus (P.M.);
Tel.: +34-637-423-096 (I.A.M.); +34-636-184-806 (P.M.)

**Abstract:** Climatic variations in the current environmental scenario require plants with tolerance to sudden changes in temperature and a decrease in water availability. Accordingly, this tolerance will enable successful plantations and the maintenance of natural and planted forests. Consequently, in the last two decades, drought tolerance and high temperatures in conifers have been an important target for morphological, physiological, and epigenetic studies. Based on this, our research team has optimized different stages of somatic embryogenesis (SE) in *Pinus* spp. improving the success of the process. Through this method, we can obtain a large amount of clonal material and then analyze the somatic plants under different conditions ex vitro. The analysis of the morphological and physiological parameters in somatic embryos (ses) and plants with different tolerances to abiotic stress can provide us with valuable information about the mechanisms used by plants to survive under adverse environmental conditions. Thus, the objective of this work was to evaluate the influence of high temperatures (23, 40, 50, and 60 °C, after 12 weeks, 90, 30, 5 min, respectively) on the morphology of somatic embryos obtained from *Pinus radiata* D.Don (Radiata pine) and *Pinus halepensis* Mill. (Aleppo pine). In addition, we carried out a physiological evaluation of the somatic plants of *P. radiata* submitted to heat and water stress in a greenhouse. We observed that the number of somatic embryos was not affected by maturation temperatures in both species. Likewise, *P. radiata* plants obtained from these somatic embryos survived drought and heat stress in the greenhouse. In addition, plants originating from embryonal masses (EMs) subjected to high maturation temperature (40 and 60 °C) had a significant increase in $g_s$ and $E$. Therefore, it is possible to modulate the characteristics of somatic plants produced by the manipulation of environmental conditions during the process of SE.

**Keywords:** abiotic stress; embryonal masses; *Pinus halepensis*; *Pinus radiata*; somatic embryogenesis; somatic plantlets; tolerance

## 1. Introduction

In the current climate change scenario, research is needed to enable plants to have greater water use efficiency in different environmental conditions [1] and to develop drought tolerance [2] and thermotolerance [3]. This thermotolerance can be achieved by changing the conditions in the different stages of somatic embryogenesis (SE) without it being necessary to enforce any change in the DNA, but only by allowing the formation of epigenetic memory [4,5].

*Pinus halepensis* is an important forest tree for reforestation in Mediterranean regions, because it has adapted to drought conditions with high temperatures [6,7]. On the other hand, *Pinus radiata* D. Don is a forest tree of economic importance due to the capacity it has for intensive use of its wood [8]. Preliminary studies with *P. radiata* species have reported that the high heat and drought tolerance in environments with climate variation is dependent on the ecotype [2,9]. The same observation has been made with Basque Country/Spain ecotypes, which are more sensitive to water stress compared to ecotypes from other parts of the world [2,9].

SE is a biotechnological tool which consists in the dedifferentiation of somatic cells and their subsequent cell re-differentiation in somatic embryos (ses), through genetic reprogramming [10]. Based on the morphogenetic response, SE is widely used to obtain large amounts of cloned material from elite material in response to an external stress stimulus [11–17].

In recent years, several studies have reported the influence of high temperature in the induction of SE in *Pinus* spp. [18–21]. In this regard, the effect of different temperatures applied in the initiation [4,19–21] and maturation [4,20,22,23] stages of SE process has been reported at 18, 23, or 28 °C. Recently, our research group has reported that somatic plants of *P. radiata* coming from EMs initiated at different temperatures have a different behavior in the greenhouse under water stress conditions [24].

Taking into account the abovementioned studies, in this work, we used higher maturation temperatures than those described in previous studies with *P. radiata* and *P. halepensis*. Thus, the objective of this work was to evaluate the influence of high temperatures (23 (control temperature), 40, 50, and 60 °C, after 12 weeks, 90, 30, 5 min, respectively) applied during the maturation stage of the SE in terms of both quantity and morphology of the ses obtained of *P. radiata* and *P. halepensis*. Moreover, we carried out a physiological evaluation of the obtained *P. radiata* plantlets subjected to heat and water stress in the greenhouse in order to test the possible improvement related to heat and water stress tolerance.

## 2. Materials and Methods

### 2.1. Experiment I

EMs obtained from immature female cones of *P. radiata*, were collected from four mother trees in a seed orchard established by Neiker-BRTA in Deba (Spain) and *P. halepensis* cones were collected from five mother trees in Berantevilla (Spain). The immature seeds were extracted and surface sterilized following Montalbán et al. [25]. Seed coats were removed and intact megagametophytes excised out aseptically were initiated and proliferated following the protocol described by Montalbán et al. for *P. radiata* [25] and for *P. halepensis* [26].

For *P. radiata* SE, the basal medium was Embryo Development Medium (EDM) (Duchefa Biochemie, Amsterdam, Netherlands) [27]. The maturation of EMs was performed following the protocol described by Montalbán et al. [28]. The EMs were briefly suspended in liquid basal medium devoiding plant growth regulators and then filtered on a filter paper in a Büchner funnel. Each filter paper, with 0.08 g of EMs, was placed in the corresponding maturation media. The basal medium was supplemented with 60 g L$^{-1}$ of sucrose, 9 g L$^{-1}$ of Gelrite$^{®}$ (Duchefa Biochemie, Amsterdam, Netherlands), 60 μM of abscisic acid (ABA), and the amino acid mixture of EDM medium used for initiation and proliferation (550 mg L$^{-1}$ of L-glutamine, 525 mg L$^{-1}$ of asparagine, 175 mg L$^{-1}$ of arginine, 19.75 mg L$^{-1}$ of citrulline, 19 mg L$^{-1}$ of ornithine, 13.75 mg L$^{-1}$ of lysine, 10 mg L$^{-1}$ of alanine, and 8.75 mg L$^{-1}$ of proline) [27].

For *P. halepensis* SE, the basal medium used was DCR medium (Duchefa Biochemie, Amsterdam, Netherlands) [29] supplemented with 75 µM of ABA, 60 g L$^{-1}$ of sucrose, 9 g L$^{-1}$ of Gelrite$^{®}$ and the amino acid mixture of the EDM medium. The maturation of the 0.08 g of EMs/ plate was carried out as described above.

The cultures were kept at different maturation temperatures (MT) (23 (control temperature), 40, 50, and 60 °C) during different incubation periods (12 weeks, 90, 30, 5 min, respectively). Based on previous studies, we observed that extreme temperatures cannot be applied over extended periods as the ECLs are killed. Once the different treatments had finished, all cultures were kept in darkness at 23 °C.

Germination and acclimatization of cotyledonary ses were performed according to Montalbán and Moncaleán [30].

Therefore, four different temperatures were tested in six established cell lines (ECLs) (R2, R9, R16, R49, R130, and R138 for *P. radiata* and H2, H23, H48, H60, H153, and H204 for *P. halepensis*), in a factorial design with eight repetitions (plates) per treatment and ECL. After 16 weeks from the beginning of the experiments, the number of normal mature somatic embryos (NNE) (for *P. radiata* and *P. halepensis*, Figure 1a,b, respectively) and abnormal somatic embryos (NAE) for *P. radiata* and *P. halepensis*, Figure 1c,d, respectively) per 0.08 g of EMs were counted. NAE displayed abnormal morphology, manifested by precocious germination, and in some cases by a lack of cotyledons [28]. Additionally, the length (LE) and width (WE) of 960 NNE was measured. After two months in germination medium, the percentage of the germination for the NNE was calculated.

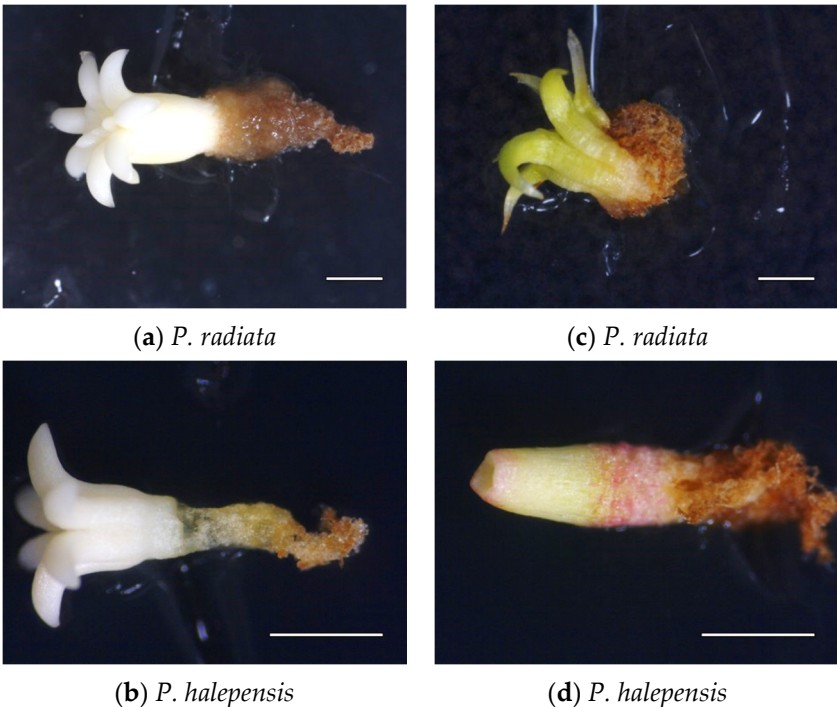

(**a**) *P. radiata*    (**c**) *P. radiata*

(**b**) *P. halepensis*    (**d**) *P. halepensis*

**Figure 1.** Somatic embryos showing distinct morphologies: (**a**) normal somatic embryo (NNE) for *Pinus radiata* D.Don; (**b**) normal somatic embryo (NNE) for *Pinus halepensis* Mill.; (**c**) abnormal somatic embryo (NAE) for *P. radiata*; (**d**) abnormal somatic embryo (NAE) for *P. halepensis*, bar = 2 mm.

*2.2. Experiment II*

Six-month-old radiata pine plants (Figure 2) were analyzed in October 2019. Twenty plants were used per MT; one fourth of each MT were randomly selected and submitted to the following four treatments: greenhouse temperature (GT) at 23 °C and under an irrigation rate of three times per week (UI); GT at 23 °C and without irrigation (NI); GT at 40 °C and UI; GT at 40 °C and NI. Irrigation conditions were maintained for two weeks, and GT at 40 °C two hours/day for five days.

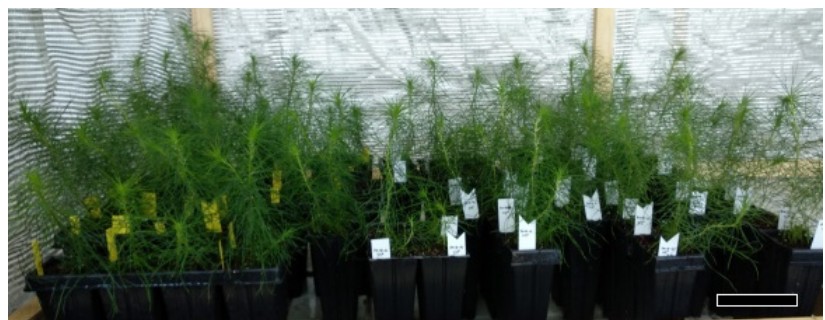

**Figure 2.** *Pinus radiata* D.Don plantlets obtained from embryonal masses maturated at high temperatures during the hardening stage in the greenhouse, bar = 8 cm.

Summarizing, in this analysis 16 treatments were considered for each species (four MT × two GT × two irrigation condition) in a factorial with five repetitions (plants), totaling 80 plants analyzed.

After two weeks from the beginning of the greenhouse experiments, plantlets from the greenhouse with a temperature at 23 °C with or without irrigation or with a greenhouse temperature at 40 °C with or without irrigation, started to present external symptoms of drought stress such as needle epinasty or apical curvature [9]. Subsequently, the leaf water status and gas exchange parameters were measured. The water potential ($\Psi_{leaf}$, MPa) of one needle per plant, was measured at predawn using a Scholander chamber (Skye SKPM 1400) and the pressure-equilibration technique [31].

The response of instant net photosynthesis ($A_N$, μmol $CO_2$ m$^{-2}$s$^{-1}$), stomatal conductance ($g_s$, mmol $H_2O$ m$^{-2}$s$^{-1}$), and instant leaf transpiration ($E$, mmol $H_2O$ m$^{-2}$s$^{-1}$) were quantified at midday with a LI-6400XT Portable Photosynthesis System (Li-Cor Biosciences) equipped with the 6400-05 Clear Conifer Chamber (Li-Cor Biosciences).

*2.3. Statistical Analysis*

For experiment I, ECLs were considered as a block in the model to decrease variability. Deviance analysis was performed with the $X^2$ test ($p < 0.05$) to assess the effect of temperature on the parameters studied. According to the data distribution, Poisson (NNE and NAE) or Gamma (LE and WE) distribution for *P. radiata*, and Poisson (NNE and NAE) and normal (LE and WE) were used for *P. halepensis*. The data were analyzed using R software®, version 3.6.1. [32] using the general linear model (glm) function.

To assess the effect of maturation temperature on the percentage of germinated NNE and the percentage of surviving plants, a logistic regression and the corresponding analysis of deviance was conducted. The cell line was included in the model to cope with variability. When required, a quasibinomial family was considered to deal with overdispersion.

For experiment II, an analysis of variance was conducted to assess the effects of MT, greenhouse temperature (GT), and irrigation condition (I) on water potential, instant net photosynthesis, stomatal conductance, and instant leaf transpiration. Full models with the complete interaction MT × GT × I were fitted.

After the analysis of variance was conducted, differences in means between treatment combinations were assessed using the Tukey post-hoc test ($\alpha = 0.05$) adjusted for multiple comparisons.

## 3. Results

### 3.1. Experiment I

#### 3.1.1. P. radiata

The application of high temperatures caused statistically significant differences for the NNE and NAE (Table 1). However, the application of high temperature was not statistically significant for the LE and WE (Table 1).

**Table 1.** Analysis of deviance for the effect of different maturation temperatures in the number of normal (NNE) and abnormal somatic embryos (NAE) per 0.08 g of embryonal masses; the length (LE-mm) and width of normal embryos (WE-mm) of *Pinus radiata* D.Don.

| Source | df | NNE | | NAE | | LE | | WE | |
|---|---|---|---|---|---|---|---|---|---|
| | | $x^2$ Test | *p*-Value | $x^2$ Test | *p*-Value | $x^2$ Test | *p*-Value | $x^2$ Test | *p*-Value |
| T | 3 | 34.69 | ≤0.05 * | 27.56 | ≤0.05 * | 0.04 | >0.05 [ns] | 0.09 | >0.05 [ns] |

* Significant differences at $p \leq 0.05$; [ns] nonsignificant; df—degrees of freedom.

There was a tendency to increase the NNE (Figure 3a) and the NAE (Figure 3a) from MT 23 °C to MT 50 °C, and the NNE decreased significantly at 60 °C.

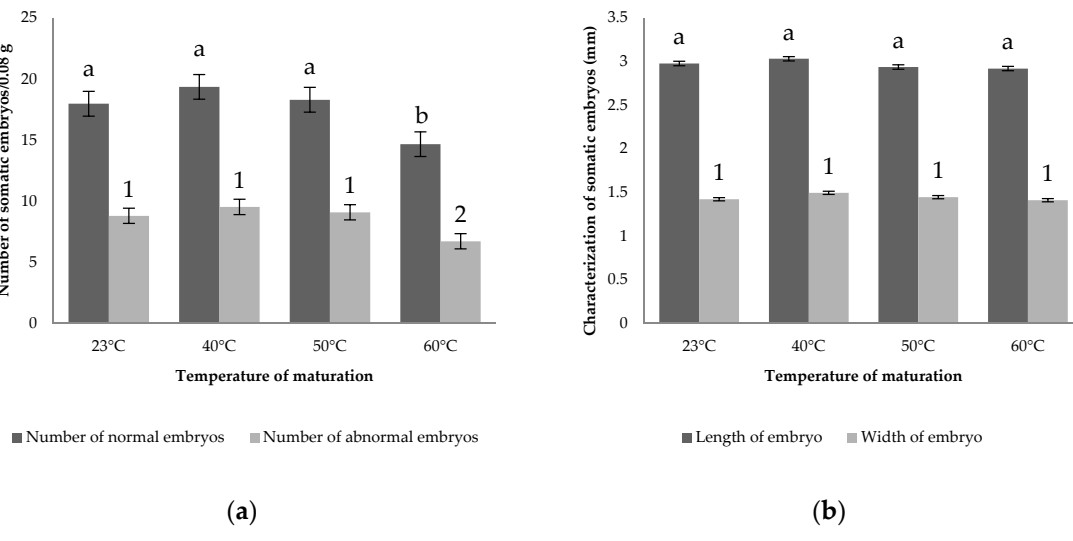

(**a**)　　　　　　　　　　　　　　　　　　　　　　　　(**b**)

**Figure 3.** Somatic embryos obtained per 0.08 g of embryonal masses submitted in different maturation temperatures (23, 40, 50, and 60 °C, after 12 weeks, 90, 30, 5 min, respectively). (**a**) Number of normal and abnormal somatic embryos; (**b**) the length and width of *Pinus radiata* D.Don normal embryos. Different letters or numbers show significant differences by the Tukey–Kramer test ($p \leq 0.05$).

Although nonsignificant differences were found between MT for LE and WE (Table 1), a similar trend was observed for NNE and NAE. The highest LE and WE were recorded in ses from MT 40 (3.03 mm for LE and 1.49 for WE) and 23 °C; the 50 °C treatment showed intermediate values and treatment at 60 °C presented the lowest (2.92 mm for LE and 1.42 mm for WE). We also observed that the longest embryos were also the widest (Figure 3b).

The application of high temperature was not statistically significant for the germination of ses and the survival percentage of somatic plants in the greenhouse (44%, 51%, 46%, and 51% for 23, 40, 50, and 60 °C, respectively). However, plantlets were obtained from all the tested maturation treatments (63%, 55%, 60%, and 63% from temperatures 23, 40, 50, and 60 °C, respectively) (Figure 4a,b) and a total of 614 plantlets were acclimated in a greenhouse (Figure 4c).

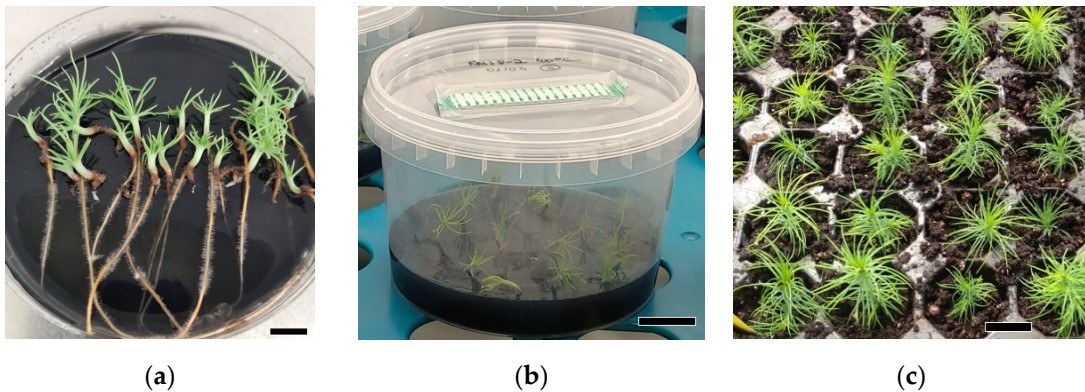

(**a**)  (**b**)  (**c**)

**Figure 4.** Germination and acclimatization of somatic plantlets from *P. radiata* obtained from different embryogenic cell lines and maturation temperatures. (**a**) Plantlets after two months in germination medium, bar = 1 cm; (**b**) plantlets after three months cultivated in the germination medium, bar = 2 cm; (**c**) plantlets derived from normal cotyledonary somatic embryos growing in the greenhouse, bar = 2 cm.

### 3.1.2. *P. halepensis*

Maturation temperature affected significantly the NNE and LE. On the contrary, MT was not statistically significant for the NAE and WE (Table 2).

**Table 2.** Analysis of deviance for the effect of different maturation temperatures in the number of normal (NNE) and aberrant somatic embryos (NAE) per 0.08 g of embryonal masses; the length (LE-mm) and width of normal embryos (WE-mm) of *Pinus halepensis* Mill.

| Source | df | NNE | | NAE | | LE | | WE | |
|---|---|---|---|---|---|---|---|---|---|
| | | $x^2$ Test | *p*-Value | $x^2$ Test | *p*-Value | $x^2$ Test | *p*-Value | $x^2$ Test | *p*-Value |
| T | 3 | 21.15 | ≤0.05 * | 0.98 | >0.05 ns | 2.44 | ≤0.05 * | 0.17 | >0.05 ns |

* Significant differences at $p \leq 0.05$; ns nonsignificant; df—degrees of freedom.

Unlike in radiata pine, the increase in the MT from 23 to 40 °C did not promote an increase in NNE, but a decrease in their production was observed (Figure 5a). On the other hand, no significant differences were found between the control MT (23 °C) and temperatures of 50 or 60 °C (Figure 5a).

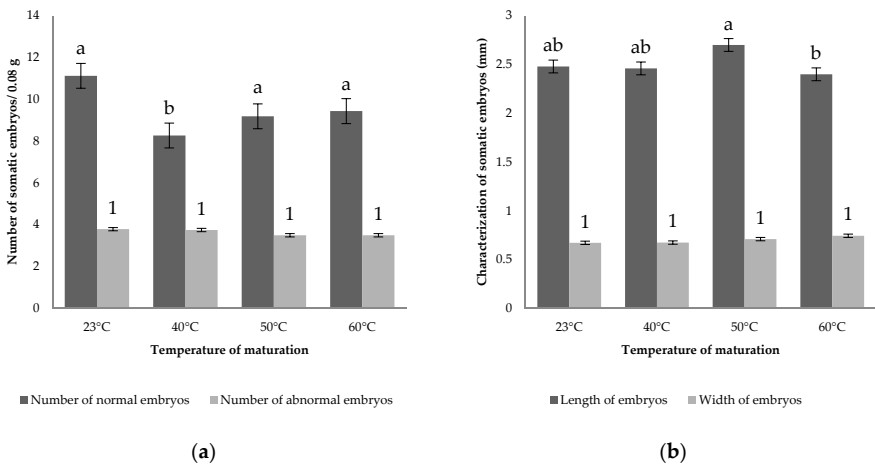

(**a**)  (**b**)

**Figure 5.** Somatic embryos obtained per 0.08 g of embryonal masses submitted in different maturation temperatures (23, 40, 50, and 60 °C, after 12 weeks, 90, 30, 5 min, respectively). (**a**) Number of normal and abnormal somatic embryos; (**b**) the length and width of normal embryos from *Pinus halepensis* Mill. Different letters or numbers show significant differences according to the Tukey–Kramer test ($p \leq 0.05$).

Although no significant differences were detected between MT for the NAE (Table 2) (Figure 5a), the MT 50 °C promoted a significant increase in LE when compared to 60 °C, whereas MT 23 and 40 °C led to intermediate values (Figure 5b). A similar tendency was observed in WE, but in this case no significant differences were found between MT (Figure 5b).

In total, 262 viable plantlets were obtained in this experiment (Figure 6c). The application of high temperature was not statistically significant for the germination percentage of ses and the survival of somatic plants in the greenhouse (22%, 30%, 24%, and 34% for 23, 40, 50, and 60 °C, respectively). Furthermore, in all ECLs and MTs, a high acclimatization rate was obtained (91.82%, 92.61%, 98.08%, and 88.91% from temperatures of 23, 40, 50, and 60 °C, respectively).

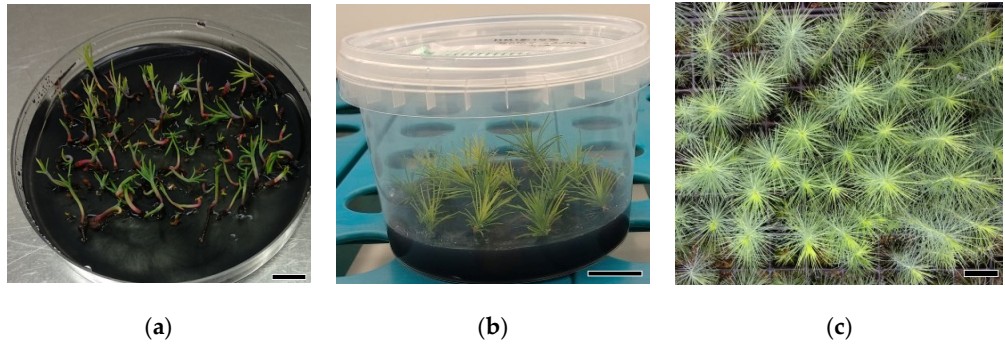

(**a**) (**b**) (**c**)

**Figure 6.** Germination and acclimatization of somatic plantlets from *Pinus halepensis* Mill. obtained from different embryogenic cell lines and maturation temperatures. (**a**) Somatic plantlets after two months in germination medium, bar = 1 cm; (**b**) plantlets cultivated after three months in the germination medium, bar = 2 cm; (**c**) plantlets derived from normal cotyledonary somatic embryos growing in the greenhouse, bar = 5 cm.

### 3.2. Experiment II

Water Potential and Gas Exchange Parameters

Plants coming from all treatments applied during the maturation stage survived after the drought and thermic stress in the greenhouse (Figure 7). Significant differences were observed for the effect of MT on the initial and final evaluation of water potential in plants subjected to different stress conditions (Table 3); but in the first evaluation no significant differences were observed for gas exchange parameters (Table 3).

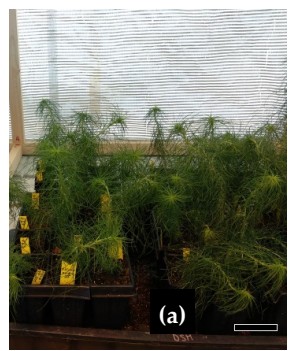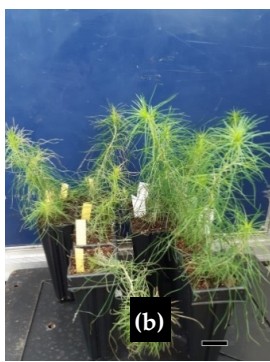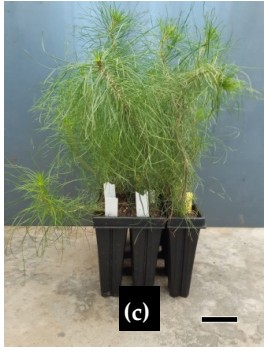

**Figure 7.** *Pinus radiata* D.Don somatic plants: (**a**) showing needle epinasty and apical curvature at greenhouse temperature of 40 °C under irrigation or nonirrigated (left and right, respectively), bar = 6 cm; (**b**) plants not irrigated submitted to 40 and 23 °C in greenhouse (left and right, respectively), bar = 3 cm; (**c**) plants not irrigated submitted to 23 and 40 °C in greenhouse (left and right, respectively), after rewatering, bar = 6 cm.

**Table 3.** Analysis of variance (ANOVA) for the effect of different maturation temperatures (MT) and greenhouse temperatures (GT) on the initial and final water potential ($\Psi_{leaf}$ initial and $\Psi_{leaf}$ final, respectively) in plants of *Pinus radiata* D.Don under irrigation and/or no irrigation conditions (I).

| ANOVA | | |
|---|---|---|
| **Effect** | **df** | **$\Psi_{leaf}$ initial** |
| MT | 3 | ≤0.05 * |
| **Effect** | **df** | **$\Psi_{leaf}$ initial** |
| MT | 3 | ≤0.05 * |
| GT | 1 | ≤0.05 * |
| I | 1 | >0.05 [ns] |
| MT × GT | 3 | >0.05 [ns] |
| MT × I | 3 | >0.05 [ns] |
| GT × I | 1 | >0.05 [ns] |
| MT × GT × I | 3 | >0.05 [ns] |

* Significant differences at $p \leq 0.05$; [ns] nonsignificant at $p \leq 0.05$; df—degrees of freedom.

Statistically significant differences were also found for GT for the final assessment of water potential (Table 3). At the end of the experiment, water potential in plants at GT of 23 °C was significantly lower (−0.3 MPa) than $\Psi_{leaf}$ final in plants at GT of 40 °C (−0.26 MPa) by Student's *t*-test ($p < 0.05$). This implies that plants originating from EMs subjected to different MTs had a tolerance to heat stress in the greenhouse, as plants exposed to GT of 40 °C were more hydrated compared to plants exposed to GT of 23 °C. The irrigation regime did not show significant differences for this parameter.

A decrease in water potential was observed in plants from ECLs matured in MTs of 23 °C (Figure 8a). Similarly, the same pattern of decline was observed when assessing water potential after water and heat stress conditions, which in turn was less than the initial water potential (Figure 8a). For this characteristic, the data fit the model with a determination coefficient above 98% ($R^2 = 0.98$).

In the first evaluation, there were no significant statistical differences for $A_N$, $g_s$, and $E$ (Table 4). By contrast, in the second evaluation, the $A_N$ presented significant statistical differences for the effects MT, MT × GT, GT × I, and in the triple interaction MT × GT × I (Table 5). Stomatal conductance was statistically significant for all the factors and instant transpiration was statistically significant for all the factors assessed, except MT × I and the triple interaction (Table 5).

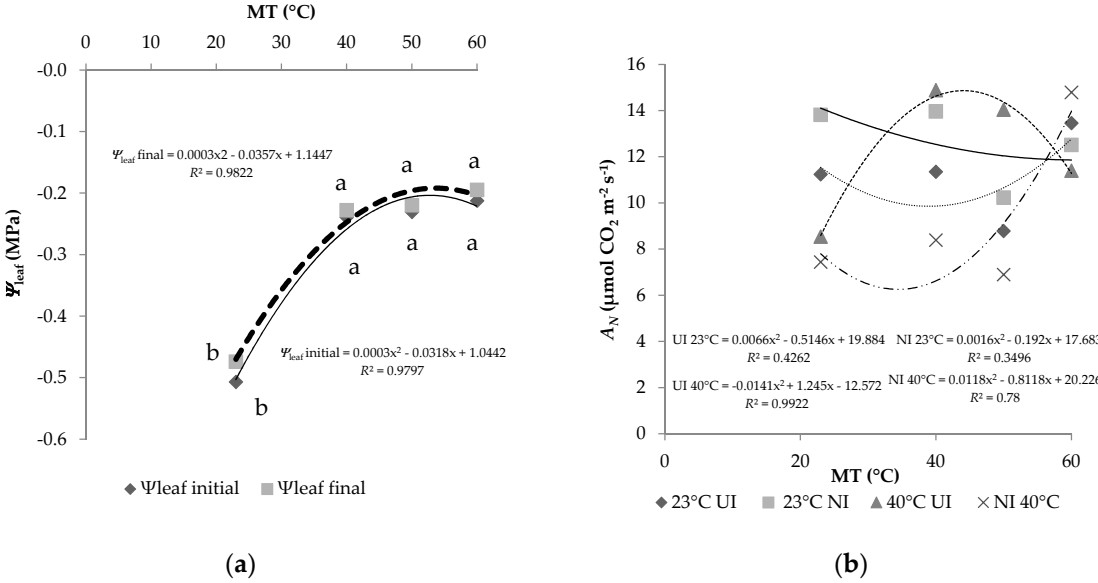

(**a**)

(**b**)

**Figure 8.** *Cont.*

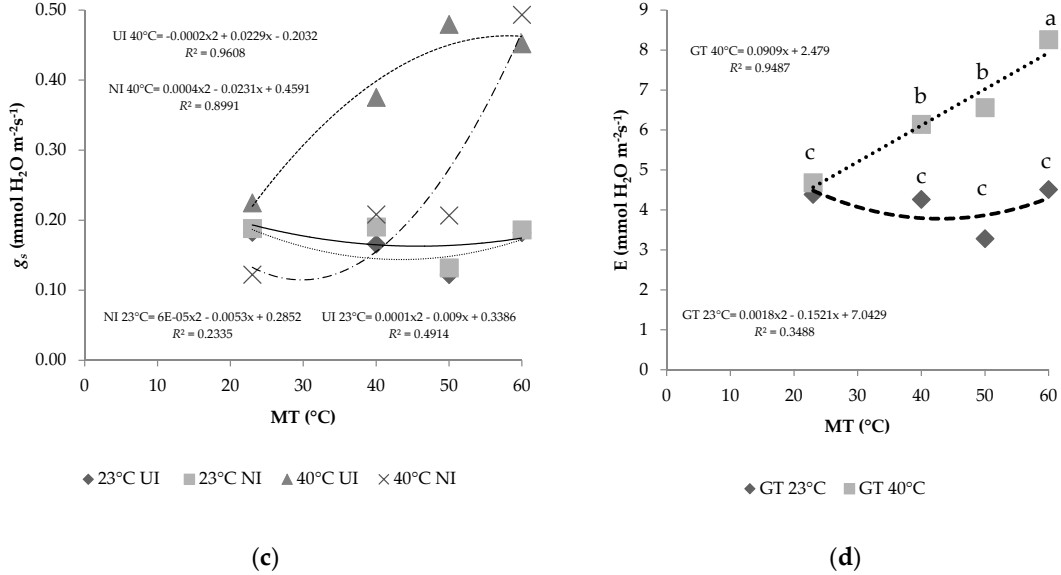

**(c)**　　　　　　　　　　　　　　　　　　　　**(d)**

**Figure 8.** Water potentials and gas exchange parameters in plants with greenhouse temperature (23 or 40 °C) under (UI) irrigation or no irrigation (NI) conditions (I) obtained from EMs of *Pinus radiata* D.Don submitted to different maturation temperatures (MT). (**a**) Leaf water potentials ($\Psi_{leaf}$) initial and final (applying of these treatments), (**b**) instantaneous net photosynthesis ($A_N$, $\mu$mol $CO_2$ m$^{-2}$ s$^{-1}$), (**c**) stomatal conductance ($g_s$, mmol $H_2O$ m$^{-2}$ s$^{-1}$), and (**d**) instant transpiration (E, mmol $H_2O$ m$^{-2}$ s$^{-1}$). Different letters show significant differences by the Tukey–Kramer test ($p \leq 0.05$).

**Table 4.** ANOVA for the effect of different maturation temperatures (MT) in the instantaneous net photosynthesis ($A_N$, $\mu$mol $CO_2$ m$^{-2}$ s$^{-1}$), stomatal conductance ($g_s$, mmol $H_2O$ m$^{-2}$s$^{-1}$), and instant transpiration (E, mmol $H_2O$ m$^{-2}$s$^{-1}$) in plants of *Pinus radiata* D.Don.

| ANOVA Gas Exchange Initial | | | | |
|---|---|---|---|---|
| **Effect** | **df** | $A_N$ | $g_s$ | $E$ |
| MT | 3 | >0.01 [ns] | >0.01 [ns] | >0.01 [ns] |

[ns] nonsignificant at $p \leq 0.05$; df—degrees of freedom.

**Table 5.** ANOVA for the effect of different maturation temperatures (MT), greenhouse temperature (GT), and irrigation or no irrigation conditions (I) in the instantaneous net photosynthesis ($A_N$, $\mu$mol $CO_2$ m$^{-2}$ s$^{-1}$), stomatal conductance ($g_s$, mmol $H_2O$ m$^{-2}$s$^{-1}$), and instant transpiration (E, mmol $H_2O$ m$^{-2}$s$^{-1}$) in plants of *Pinus radiata* D.Don.

| ANOVA Gas Exchange Final | | | | |
|---|---|---|---|---|
| **Effect** | **df** | $A_N$ | $g_s$ | $E$ |
| MT | 3 | ≤0.01 ** | ≤0.001 *** | ≤0.001 *** |
| GT | 1 | >0.01 [ns] | ≤0.001 *** | ≤0.001 *** |
| I | 1 | >0.01 [ns] | ≤0.01 ** | ≤0.01 ** |
| MT × GT | 3 | ≤0.05 * | ≤0.001 *** | ≤0.001 *** |
| MT × I | 3 | >0.01 [ns] | ≤0.01 ** | >0.01 [ns] |
| GT × I | 1 | ≤0.01 ** | ≤0.001 *** | ≤0.001 *** |
| MT × GT × I | 3 | ≤0.01 ** | ≤0.01 ** | >0.01 [ns] |

*; **; *** Significant differences at $p \leq 0.05$, $p \leq 0.01$, or $p \leq 0.001$, respectively; [ns] nonsignificant at $p \leq 0.01$; df—degrees of freedom.

As we can see in Table 6, the relative effect of heat and water stress did not induce statistically significant differences in instantaneous liquid photosynthesis between any treatments, except those plants originating from ECLs submitted to maturation temperatures of 23 and 50 °C (Figure 8b).

**Table 6.** Effect of different maturation temperatures (MT-°C), greenhouse temperature (GT-°C), and irrigation (1) or no irrigation (2) conditions (I) on the instantaneous net photosynthesis ($A_N$, μmol $CO_2$ m$^{-2}$ s$^{-1}$), stomatal conductance ($g_s$, mmol $H_2O$ m$^{-2}$s$^{-1}$) in plants of *Pinus radiata* D.Don.

| Effect (MT-GT-I) | $A_N$ | $g_s$ |
|---|---|---|
| 40-40-1 | 14.89 [a] | 0.38 [b,c] |
| 60-40-2 | 14.79 [a] | 0.49 [a] |
| 50-40-1 | 14.04 [a] | 0.48 [a,b] |
| 40-23-2 | 13.97 [a] | 0.19 [c,d] |
| 23-23-2 | 13.82 [a] | 0.19 [c,d] |
| 60-23-1 | 13.46 [a] | 0.18 [c,d] |
| 60-23-2 | 12.52 [a] | 0.19 [c,d] |
| 60-40-1 | 11.40 [a] | 0.45 [a,b] |
| 40-23-1 | 11.35 [a] | 0.17 [c,d] |
| 23-23-1 | 11.24 [a] | 0.18 [c,d] |
| 50-23-2 | 10.24 [a] | 0.13 [d] |
| 50-23-1 | 8.78 [a] | 0.12 [d] |
| 23-40-1 | 8.54 [a] | 0.23 [c,d] |
| 40-40-2 | 8.39 [a] | 0.21 [c,d] |
| 23-40-2 | 7.44 [b] | 0.12 [d] |
| 50-40-2 | 6.90 [b] | 0.21 [c,d] |

Different letters within a column show significant differences in the means observed by Tukey–Kramer's post hoc test ($p \leq 0.05$).

However, plants have greater variability with statistically significant differences in stomatal conductance (Table 6). In addition, the plants that maintained higher stomatal conductance compared to those under stress conditions or otherwise, were plants originating from ECLs exposed to high maturation temperatures (0.49 mmol $H_2O$ m$^{-2}$s$^{-1}$ by 60 °C) (Figure 8c) (Table 6).

At the end of the experiment, under high temperature conditions in a greenhouse, we observed a significantly increase in the instant transpiration in plants originating from ECLs subjected to MT at 60 °C, followed by 50 and 40 °C. We also observed that this increase was greater under conditions of heat stress than in nonstressed plants (Figure 8d). This parameter also showed statistically significant differences in plants growing in the greenhouse at 40 °C under irrigation, with an increase in transpiration in this cultivation condition (Table 7).

**Table 7.** Effect of different greenhouse temperatures (GT, °C) and irrigation (UI) or no irrigation (NI) conditions (I) in the instant transpiration (E, mmol $H_2O$ m$^{-2}$ s$^{-1}$) in plants of *Pinus radiata* D.Don.

| Effect (GT (°C)-I) | E |
|---|---|
| 40-UI | 7.19 [a] |
| 40-NI | 5.61 [b] |
| 23-NI | 4.26 [b,c] |
| 23-UI | 3.96 [c] |

Different letters show significant differences in the means observed by Tukey–Kramer's post hoc test ($p \leq 0.05$).

## 4. Discussion

The formation of somatic embryos is affected by several factors, such the genetic background and the cultivation conditions [33,34]. In this study, the same ECLs (five for each species), were subjected to different maturation treatments, in order to try to minimize the genetic impact on the results obtained. In our experiments we observed that, in both species, embryos were formed with normal and abnormal morphologies. Merino et al. [35] in Scots pine attributed the formation of ses with normal and abnormal morphology to the difference of transcripts in embryogenic cells. However, the increase in temperature did not significantly promote the formation of abnormal embryos.

In this work we observed that the NNE was not affected by high temperatures applied at the beginning of maturation stage in *P. radiata*, except in *P. halepensis* in which we observed a reduction in

NNE in at 50 °C. Kvaalen and Johnsen [22] and García-Mendiguren et al. [18], when working with *Picea abies* and *P. radiata*, respectively, also reported an increase in the number of embryos obtained when high temperatures were applied in the initiation stage. In contrast, Arrillaga et al. [20], in proliferation and maturation stages *Pinus pinaster* SE showed that control temperature (23 °C) provoked the best results in terms of the number of embryos developed.

Castander-Olarieta et al. [21] in *P. radiata* showed that somatic embryos originating from EMs initiated at 50 °C for 30 min had a higher LE. This is in agreement with our results for Aleppo pine where the longest embryos were found at this temperature.

The germination mechanism of cotyledonary somatic embryos is complex and an alternative in this case are studies with post-maturation treatments [36], because in *P. halepensis* (22%, 30%, 24%, and 34% for 23, 40, 50, and 60 °C, respectively) we observed a low percentage rate of germination compared to *P. radiata* (44%, 51%, 46%, and 51% for 23, 40, 50, and 60 °C, respectively).

Stresses applied to EMs in late SE stages, besides inducing different responses between species of conifers, can also guarantee an improvement in the germination of cotyledonary somatic embryos [37]. The germination rate changes according to species and treatments. This result was also described in other conifer species [20,38,39]. In this work, we observed that the high MTs did not affect either the germination of ses or the acclimatization of plants in both species of *Pinus*.

The conditions of cultivation during the maturation stage in the SE are crucial for the regeneration of the somatic embryos in quality plantlets with an ex vitro survival capacity [40]. We obtained the highest acclimatization (≤88.91%) independently of the line or temperature for *P. halepensis* in relation to *P. radiata* (ranging from 55.84% to 63.47% for temperature). The maturation of EMs in ses and their subsequent conversion to plants, in both species, prove the tolerance of these EMs to high temperatures.

The pressures exerted by high temperatures and drought affect numerous defense mechanisms in plants [41], such as changes in photosynthetic machinery [42]. This can happen at an embryogenic level with the formation of an epigenetic memory [4] even at the plant level. In this work, we observed that the different MTs applied to EMs promoted ses without significant physical anomalies. However, *P. radiata* plants originated from these treatments responded physiologically in different ways (Figure 8) when they were subjected to drought and heat stress in the greenhouse.

Taiz and Zeiger [43] reported that water stress significantly affects stomatal conductance more than photosynthesis, corroborating our results (Table 6). In addition, we also observed that $E$ was affected by heat and water stress (Figure 8d and Table 7).

It is possible that plants originating from ECLs subjected to high temperatures in the maturation stage have developed epigenetic mechanisms that allowed a better response to stress [44], since the plants maintained or adapted their water and gas exchange potential to adverse temperature conditions and water stress to which they were subjected (Figure 8).

In this work, plants from all MT submitted to GT of 23 °C showed a similar behavior for $g_s$ and $E$. However, when subjected to heat stress (40 °C), plants originating from EMs subjected to high MTs (40 and 60 °C) had a significant increase in $g_s$ and $E$, reinforcing once again the possibility of "priming" in ECLs subjected to high temperatures. In natural conditions, similar behavior is observed in the apical meristem of plants, which is the center of the morphogenesis of the aerial part and is in constant cell division. The environmental restrictions, for example, heat and drought, perceived by these meristematic regions, trigger changes in the epigenetic state of the plants, developing a memory which, when under stress conditions again, enables the plants to have better tolerance [45].

In conclusion, it is possible to modulate the tolerance to stress by applying high temperatures during the final stages of the embryogenic process. Future experiments will be carried out to analyze methylation, physiological and biochemical aspects in one-year old plants to assess whether the plant characteristics endure over time.

## 5. Conclusions

*P. radiata* and *P. halepensis* EMs produced ses at high maturation temperatures. High maturation temperatures compared to the control temperature (23 °C) did not affect the morphological characteristics of the embryos obtained, except for the LE in both species, and WE in *P. halepensis*. The plants obtained from these somatic embryos survived drought and heat stresses in the greenhouse. Moreover, plants originated from EMs submitted to a maturation temperature of 40 and 60 °C, presented better adaptation to drought and heat stress based on the water potential and gas exchange parameters analyzed in this experiment. Studies will be carried out to characterize possible epigenetic marks in the obtained plants caused by heat stress applied during the embryogenic process and drought and heat stress applied in plants in greenhouses.

**Author Contributions:** P.M. and I.A.M. conceived and planned the experiments. A.M.M.d.N. performed the experiments. P.A.B., N.F.F.d.N., T.G. and M.D.U. carried out the statistical analyses. A.M.M.d.N. wrote the manuscript and all authors provided critical feedback and helped shape the manuscript. All authors have read and agreed to the published version of the manuscript.

**Funding:** This research was funded by MINECO (Spanish Government) project (AGL2016-76143-C4-3R), CYTED (P117RT0522), and MINECO (BES-2017-081249, "Ayudas para contratos predoctorales para la formación de doctores"). MULTIFOREVER (Project MULTIFOREVER) is supported under the umbrella of ERA-NET cofund Forest Value by ANR (FR), FNR (DE), MINCyT (AR), MINECO-AEI (ES), MMM (FI), and VINNOVA (SE). Forest value has received funding from the European Union's Horizon 2020 research and innovation programmed under agreement No 773324.

**Conflicts of Interest:** The authors declare no conflict of interest.

## Abbreviations

ABA, abscisic acid; $A_N$, instant net photosynthesis; $E$, instant leaf transpiration; ECLs, established cell lines; EDM, Embryo Development Medium; EMs, embryonal masses; $g_S$, stomatal conductance; GT, greenhouse's temperature; I, irrigation condition; LE, length of somatic embryo; MT, maturation temperatures; NAE, abnormal somatic embryos; NI, without irrigation; NNE, number of normal mature somatic embryos; SE, somatic embryogenesis; ses, somatic embryos; UI, under irrigation; WE, width of somatic embryo; $\Psi_{leaf}$, water potential.

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
