# Peer review of "Pinus spp. Somatic Embryo Conversion under High Temperature: Effect on the Morphological and Physiological Characteristics of Plantlets"

_forests, doi:10.3390/f11111181_

Round 1

Reviewer 1 Report

Comments to the manuscript

The authors used the experimental system of somatic embryogenesis for the study of stress response  in regenerated plantlets (somatic seedlings) exerted during maturation phase of somatic embryos in two conifer species.

This approach is novel (at least for conifer somatic embryogenesis) and can contribute to theoretical aspects of somatic embryogenesis (later maybe also  for practical exploitation  of the process). The obtained results are interesting for scientific readership.

The manuscript is recommended for publication with minor revision.

Moderate changes are recommended:

- the morphology of „normal“ as well as „abnormal“ somatic embryos is demonstrated on figures, still it woulde be useful to describe it shortly in the text (e.g in Results).

Line 71, instead of ripening the word „maturation would be better

Line 73 „high temperatures (23, 40, 50 and 60°C..) Is 23 °C high temperature?

Material and methods

Two different basal media were used for the two different species. It is recommended to give the reason shortly in discussion.

Line 161.... in the text „temperature at 23°C or without irrigation“, .....with or without?

Line 343 ....MTs did not  affected.., did not affect?

Results

The authors used different cell lines for both species. It is supposed, for the study of morphological as well as physiological parameters  plantlets  regenerated from different cell lines were used. It  is recommended to mention this approach shortly in discussion. In discussion the authors stressed the genetic factor.

Line 202 „Figure 3. Somatic embryos obtained per 0.08 g of embryonal masses“....on the axis „y“ NNE/g or NAE/g is given (1 g?), the same in Fig. 4

References

-Line 419..plantation in spain? Spain?...

Moreover, in References many Latin (scientific) names of species are written with small (lower case) letters. Is it an intention or typing error ? See  lines 412, 430, 438,444,447,449, 451, 470, 479,  etc.

Author Response

Response to Reviewer 1 Comments

Comments and Suggestions for Authors

Comments to the manuscript

The authors used the experimental system of somatic embryogenesis for the study of stress response  in regenerated plantlets (somatic seedlings) exerted during maturation phase of somatic embryos in two conifer species.

This approach is novel (at least for conifer somatic embryogenesis) and can contribute to theoretical aspects of somatic embryogenesis (later maybe also  for practical exploitation  of the process). The obtained results are interesting for scientific readership.

The manuscript is recommended for publication with minor revision.

Moderate changes are recommended:

  1. the morphology of „normal“ as well as „abnormal“ somatic embryos is demonstrated on figures, still it woulde be useful to describe it shortly in the text (e.g in Results).

Thank you for your suggestion. We have added the description of the abnormal embryos in the Material and Methods (lines: 109, 110 and 111) according to our results and as described by Montalbán et al., (2010).

Montalbán, I.A.; De Diego, N.; Moncaleán, P. Bottlenecks in Pinus radiata somatic embryogenesis: Improving maturation and germination. Trees 2010, 24, 1061-1071.

  1. Line 71, instead of ripening the word „maturation would be better

We thank you for your suggestion and have changed the text accordingly (line 71).

  1. Line 73 „high temperatures (23, 40, 50 and 60°C..) Is 23 °C high temperature?

We added it as part of the treatments because it is the control temperature used in our experiments. Then, to make it clearer in the text, we also specify that it is the control temperature (lines: 30, 73 and 99).

Material and methods

  1. Two different basal media were used for the two different species. It is recommended to give the reason shortly in discussion.

The use of different culture media has been established in our laboratory in previous studies, because the two reference species have different requirements regarding embryogenic competence.

Montalbán, I.A.; De Diego, N.; Moncaleán, P. Bottlenecks in Pinus radiata somatic embryogenesis: Improving maturation and germination. Trees 2010, 24, 1061-1071.

Pereira, C.; Montalbán, I.A.; Goicoa, T.; Ugarte, M.D.; Correia, S.; Canhoto, J.M.; Moncaleán, P. Short communication: The effect of changing temperature and agar concentration at proliferation stage in the final success of aleppo pine somatic embryogenesis. Forest Systems 2018, 26.

  1. Line 161.... in the text „temperature at 23°C or without irrigation“, .....with or without?

We thank you for your suggestion and have changed the text accordingly (line 128).

  1. Line 343 ....MTs did not  affected.., did not affect?

Thank you for spotting this error which we have corrected (line 315).

Results

  1. The authors used different cell lines for both species. It is supposed, for the study of morphological as well as physiological parameters  plantlets  regenerated from different cell lines were used. It  is recommended to mention this approach shortly in discussion. In discussion the authors stressed the genetic factor.

Thank you for your suggestion. We have extended the discussion accordingly (lines: 285, 286 and 287).

  1. Line 202 „Figure 3. Somatic embryos obtained per 0.08 g of embryonal masses“....on the axis „y“ NNE/g or NAE/g is given (1 g?), the same in Fig. 4

Thank you for your observation. The text has been changed (Figure 3 and Figure 5).

References

  1. -Line 419..plantation in spain? Spain?...

Thank you spotting this error which we have corrected.

  1. Moreover, in References many Latin (scientific) names of species are written with small (lower case) letters. Is it an intention or typing error ? See  lines 412, 430, 438,444,447,449, 451, 470, 479,  etc.

Thank you spotting this error which we have corrected.

Reviewer 2 Report

- Major Compulsory Revisions

1) The main topic of the criticism is that this manuscript (Ms) contains misprints, mistakes in English grammar and in the writing style. I recommend that the authors should use some help of a native English speaker or send the Ms to an English Editing Service that proofreads scientific writing. It is very difficult to understand different parts of the manuscript in its present form.

2) Authors should present the data in a more understandable way, for example

  1. a) use histograms for Figure 3, 5.
  2. b) improve legends for figures, e.g. explain all abbreviations, do not use abbreviations in Fig. 3a,3b,3c,3d, 5a,5b,5c,5d.
  3. c) in my opinion it is better to present tables (Table 2,3,4,5,6) either in the text or in additional materials.

3) As I understand that number of normal somatic embryos at temperature 60oC were higher compared with number of normal somatic embryos at temperature 40 and 50oC? Why? What about the number of viable embryos after these treatments?

4) Line 29-30: “23, 40, 50 and 60oC, after 12 weeks, 90, 30, 5 min, respectively”. Did I understand correctly that somatic embryos were treated

  1. a) 23oC for 12 weeks;
  2. b) 40oC for 90 min;
  3. c) 50oC for 30;
  4. d) 60oC for 5 min.

If so, why are different time points used

- Minor Compulsory Revisions

Line 35: “Ems” – explain used abbreviation.

Line 108-130: there are some problems with the Ms text.

Line 284, 287: “(Error! Reference source not found.)” What is it?

Line 62: in my opinion it is better to use “SEs”, not “ses”

Line 96: “DCR medium” – explain used abbreviation.

Author Response

Response to Reviewer 2 Comments

Comments and Suggestions for Authors

Major Compulsory Revisions

1) The main topic of the criticism is that this manuscript (Ms) contains misprints, mistakes in English grammar and in the writing style. I recommend that the authors should use some help of a native English speaker or send the Ms to an English Editing Service that proofreads scientific writing. It is very difficult to understand different parts of the manuscript in its present form.

Thank you for your suggestion. The paper has since been revised by a native English speaker with extensive experience in the revision and proofreading of academic texts.

2) Authors should present the data in a more understandable way, for example

  1. a) use histograms for Figure 3, 5.

Thank you for your suggestion. These data are now depicted in histograms (Figure 3 and Figure 5).

  1. b) improve legends for figures, e.g. explain all abbreviations, do not use abbreviations in Fig. 3a,3b,3c,3d, 5a,5b,5c,5d.

Thank you for your advice which we have followed accordingly.

  1. c) in my opinion it is better to present tables (Table 2,3,4,5,6) either in the text or in additional materials.

Thank you for your suggestion. The tables are now presented in the text.

3) As I understand that number of normal somatic embryos at temperature 60oC were higher compared with number of normal somatic embryos at temperature 40 and 50oC? Why? What about the number of viable embryos after these treatments?

In P. radiata the number of normal somatic embryos (NNE) was significantly lower at 60˚C (as can be seen in Figure 3a). In P. halepensis there are no significant differences between 23, 50 and 60˚C, for which the NNE was significantly lower at 40˚C (as we can see in Figure 5a). The NNE corresponds to the number of viable embryos.

4) Line 29-30: “23, 40, 50 and 60oC, after 12 weeks, 90, 30, 5 min, respectively”. Did I understand correctly that somatic embryos were treated

  1. a) 23oC for 12 weeks;
  2. b) 40oC for 90 min;
  3. c) 50oC for 30;
  4. d) 60oC for 5 min.

If so, why are different time points used

You understood correctly. Based on previous studies, we observed that extreme temperatures cannot be applied over extended periods as the ECLs are killed.

- Minor Compulsory Revisions

5) Line 35: “Ems” – explain used abbreviation.

Thank you for your suggestion. We have done so (lines: 35 and 36).

6) Line 108-130: there are some problems with the Ms text.

Thank you for your observation. The text has been corrected (lines: 112 and 113).

7) Line 284, 287: “(Error! Reference source not found.)” What is it?

Thank you for your observation. The reference has been corrected.

8) Line 62: in my opinion it is better to use “SEs”, not “ses”

Thank you for your suggestion. However, in this case, to avoid confusion on the part of the reader, we have not used “SEs” as “SE” already corresponds to somatic embryogenesis.

9) Line 96: “DCR medium” – explain used abbreviation.

This abbreviation was previously described by the formulator of the medium, as we can read in Gupta and Durzan (1985).

Gupta, P.K.; Durzan, D.J. Shoot multiplication from mature trees of douglas-fir (Pseudotsuga menziesii) and sugar pine (Pinus lambertiana). Plant Cell Reports 1985, 4, 177-179.

Reviewer 3 Report

Comments to the Authors,

The submitted manuscript describes the Pinus spp. somatic embryo conversion under high temperature and its effect on morphological and physiological characteristics of plantlets. The authors performed the experiments in order to characterize the influence of different high temperatures on the morphology of Pinus radiata and Pinus halepensis somatic embryos. In addition, the analysis of the effect of the heat and water stress on Pinus radiata plants in a greenhouse was performed. The manuscript reports useful data, however some of its parts have to been improved.

Specific points which require revision by the authors are as follows:

Abstract

Line 30: Please provide the full names of P. radiata and P. halepensis

Line 35: Even if the Abbreviations list is provided in the manuscript, please add the full name for “EMs”

Introduction

Line 54-56: Please provide the full names of Pinus radiata and Pinus halepensis and consequently use P. radiata and P. halepensis in the whole text.

Materials and Methods

Line 104-107: How many independent experiments were done for each treatment? It is not clear if eight plates represent one independent experiment?

Line 107-130: There is a problem with the PDF formatting of the manuscript.

Line 147: Figure 1 – The images are quite dark. Is there any possibility to provide brighter pictures? As the authors introduced already in the text the abbreviation NNE (normal mature somatic embryos), it will be very useful to add this information as well on the images, together with the NAE (abnormal somatic embryos). In addition. the Figure will be more reader-friendly when the image (a) will be composed with the image (c) as the upper panel, and the image (b) together with the image (d) as a bottom panel. Also, adding the name of the species - P. radiata and P. halepensis on the figure would greatly help for the potential readers.

Results

Figure 3 and 4: Please add the NNE to the graph 3 (c) and (d) – NNE LE (mm) and NNE WE (mm).

Figure 8 (a): after what kind of treatment – UI or NI, GT 23ºC or GT 40ºC - the initial and final leaf water potential was measured?

Line 284 and 287: There is problem with PDF formatting (“Errors”).

Author Response

Response to Reviewer 3 Comments

Comments and Suggestions for Authors

Specific points which require revision by the authors are as follows:

Abstract

  • Line 30: Please provide the full names of  radiataand P. halepensis

Thank you for your suggestion. We have done so (lines 31).

  • Line 35: Even if the Abbreviations list is provided in the manuscript, please add the full name for “EMs”

Thank you for your suggestion. This has now been done (lines: 35 and 36).

Introduction

  • Line 54-56: Please provide the full names of Pinus radiataand Pinus halepensis and consequently use  radiata and P. halepensis in the whole text.

Thank you. We have followed your suggestion.

Materials and Methods

  • Line 104-107: How many independent experiments were done for each treatment? It is not clear if eight plates represent one independent experiment?

The experiment was carried out by testing different maturation temperatures. For this, we used two species of Pinus, four maturation temperatures. Each treatment (ECLsxMT) was composed of eight replicates (plate). On each plate we placed 0.08 g of ECLs.

  • Line 107-130: There is a problem with the PDF formatting of the manuscript.

Thank you for your observation. The formatting issue has been resolved.

  • Line 147: Figure 1– The images are quite dark. Is there any possibility to provide brighter pictures? As the authors introduced already in the text the abbreviation NNE (normal mature somatic embryos), it will be very useful to add this information as well on the images, together with the NAE (abnormal somatic embryos). In addition. the Figure will be more reader-friendly when the image (a) will be composed with the image (c) as the upper panel, and the image (b) together with the image (d) as a bottom panel. Also, adding the name of the species -  radiata and P. halepensis on the figure would greatly help for the potential readers.

Thank you for your suggestions. We have endeavored to improve the images and the accompanying text.

Results

  • Figure 3 and 4: Please add the NNE to the graph 3 (c) and (d) – NNE LE (mm) and NNE WE (mm).

We appreciate your suggestion. We also changed the graph following a comment from Reviewer 2.

  • Figure 8 (a): after what kind of treatment – UI or NI, GT 23ºC or GT 40ºC - the initial and final leaf water potential was measured?

As mentioned in the Results section no significant differences were found for irrigation or temperature regimes in the greenhouse. Therefore an only value for the different MT has been displayed in Figure 8a.

  • Line 284 and 287: There is problem with PDF formatting (“Errors”).

Thank you for your observation. The formatting has been corrected

Round 2

Reviewer 2 Report

1) This manuscript (Ms) still contains misprints, mistakes in English grammar and in the writing style. I recommend that the authors should use some help of a native English speaker or send the Ms to an English Editing Service that proofreads scientific writing.

2) Authors should verify statistical treatment presented in Fig. 5a – 40 and 50oC really statistically differed?

3) “As I understand that number of normal somatic embryos at temperature 60oC were higher compared with number of normal somatic embryos at temperature 40 and 50oC? Why? What about the number of viable embryos after these treatments?

Answer: You understood correctly. Based on previous studies, we observed that extreme temperatures cannot be applied over extended periods as the ECLs are killed.”

Please, include this information in Ms text.

Author Response

Response to Reviewer 2 Comments

Comments and Suggestions for Authors

  • This manuscript (Ms) still contains misprints, mistakes in English grammar and in the writing style. I recommend that the authors should use some help of a native English speaker or send the Ms to an English Editing Service that proofreads scientific writing.

Thank you for your suggestion. The paper has since been revised (twice) again by a native English speaker with extensive experience in the revision and proofreading of academic texts [Steve Hare from McDonnell English Services SL (+34 945233341/+34 665722553; www.mcdonnellenglish.com)].

  • Authors should verify statistical treatment presented in Fig. 5a – 40 and 50oC really statistically differed?

Yes. We carefully reviewed the data obtained and confirmed that for NNE were found significant statistical differences between 40 °C and 50 °C (this maturation temperature was statistically equal without significant differences with 23 °C and 60 °C).

  • “As I understand that number of normal somatic embryos at temperature 60oC were higher compared with number of normal somatic embryos at temperature 40 and 50oC? Why? What about the number of viable embryos after these treatments?

Answer: You understood correctly. Based on previous studies, we observed that extreme temperatures cannot be applied over extended periods as the ECLs are killed.”

Please, include this information in Ms text.

Thank you for your suggestion. We have done so (lines: 100, 101 and 102).